# The Consistency of Common Neighbors for Link Prediction in Stochastic Blockmodels

**Purnamrita Sarkar**
Department of Statistics
University of Texas at Austin
purnamritas@austin.utexas.edu

**Deepayan Chakrabarti**
IROM, McCombs School of Business
University of Texas at Austin
deepay@utexas.edu

**Peter Bickel**
Department of Statistics
University of California, Berkeley
bickel@stat.berkeley.edu

## Abstract

Link prediction and clustering are key problems for network-structured data. While spectral clustering has strong theoretical guarantees under the popular stochastic blockmodel formulation of networks, it can be expensive for large graphs. On the other hand, the heuristic of predicting links to nodes that share the most common neighbors with the query node is much fast, and works very well in practice. We show theoretically that the common neighbors heuristic can extract clusters with high probability when the graph is dense enough, and can do so even in sparser graphs with the addition of a "cleaning" step. Empirical results on simulated and real-world data support our conclusions.

## 1 Introduction

Networks are the simplest representation of relationships between entities, and as such have attracted significant attention recently. Their applicability ranges from social networks such as Facebook, to collaboration networks of researchers, citation networks of papers, trust networks such as Epinions, and so on. Common applications on such data include ranking, recommendation, and user segmentation, which have seen wide use in industry. Most of these applications can be framed in terms of two problems: (a) *link prediction*, where the goal is to find a few similar nodes to a given query node, and (b) *clustering*, where we want to find groups of similar individuals, either around a given seed node or a full partitioning of all nodes in the network.

An appealing model of networks is the stochastic blockmodel, which posits the existence of a latent cluster for each node, with link probabilities between nodes being simply functions of their clusters. Inference of the latent clusters allows one to solve both the link prediction problem and the clustering problem (predict all nodes in the query node's cluster). Strong theoretical and empirical results have been achieved by *spectral clustering*, which uses the singular value decomposition of the network followed by a clustering step on the eigenvectors to determine the latent clusters.

However, singular value decomposition can be expensive, particularly for (a) large graphs, when (b) many eigenvectors are desired. Unfortunately, both of these are common requirements. Instead, many fast heuristic methods are often used, and are empirically observed to yield good results [8]. One particularly common and effective method is to predict links to nodes that share many "common neighbors" with the query node $q$, i.e., rank nodes by $|CN(q,i)|$, where $CN(q,i) = \{u \mid q \sim u \sim i\}$ ($i \sim j$ represents an edge between $i$

and $j$). The intuition is that $q$ probably has many links with others in its cluster, and hence probably also shares many common friends with others in its cluster. Counting common neighbors is particularly fast (it is a JOIN operation supported by all databases and Map-Reduce systems). In this paper, we study the theoretical properties of the common neighbors heuristic.

Our contributions are the following:

(a) We present, to our knowledge the first, theoretical analysis of the common neighbors for the stochastic blockmodel.

(b) We demarcate two regimes, which we call semi-dense and semi-sparse, under which common neighbors can be successfully used for both link prediction and clustering.

(c) In particular, in the semi-dense regime, the number of common neighbors between the query node $q$ and another node within its cluster is significantly higher than that with a node outside its cluster. Hence, a simple threshold on the number of common neighbors suffices for both link prediction and clustering.

(d) However, in the semi-sparse regime, there are too few common neighbors with any node, and hence the heuristic does not work. However, we show that with a simple additional "cleaning" step, we regain the theoretical properties shown for the semi-dense case.

(e) We empirically demonstrate the effectiveness of counting common neighbors followed by the "cleaning" post-process on a variety of simulated and real-world datasets.

## 2   Related Work

Link prediction has recently attracted a lot of attention, because of its relevance to important practical problems like recommendation systems, predicting future connections in friendship networks, better understanding of evolution of complex networks, study of missing or partial information in networks, etc [9, 8]. Algorithms for link prediction fall into two main groups: similarity-based, and model-based.

**Similarity-based methods:** These methods use similarity measures based on network topology for link prediction. Some methods look at nodes two hops away from the query node: counting common neighbors, the Jaccard index, the Adamic-Adar score [1] etc. More complex methods include nodes farther away, such as the Katz score [7], and methods based on random walks [16, 2]. These are often intuitive, easily implemented, and fast, but they typically lack theoretical guarantees.

**Model-based methods:** The second approach estimates parametric models for predicting links. Many popular network models fall in the latent variable model category [12, 3]. These models assign $n$ latent random variables $Z := (Z_1, Z_2, \dots, Z_n)$ to $n$ nodes in a network. These variables take values in a general space $\mathcal{Z}$. The probability of linkage between two nodes is specified via a symmetric map $h : \mathcal{Z} \times \mathcal{Z} \to [0, 1]$. These $Z_i$'s can be i.i.d Uniform(0,1) [3], or positions in some $d-$dimensional latent space [12]. In [5] a mixture of multivariate Gaussian distributions is used, each for a separate cluster. A Stochastic Blockmodel [6] is a special class of these models, where $Z_i$ is a binary length $k$ vector encoding membership of a node in a cluster. In a well known special case (the planted partition model), all nodes in the same cluster connect to each other with probability $\alpha$, whereas all pairs in different clusters connect with probability $\gamma$. In fact, under broad parameter regimes, the blockmodel approximation of networks has recently been shown to be analogous to the use of histograms as non-parametric summaries of an unknown probability distribution [11]. Varying the number of bins or the bandwidth corresponds to varying the number or size of communities. Thus blockmodels can be used to approximate more complex models (under broad smoothness conditions) if the number of blocks are allowed to increase with $n$.

**Empirical results:** As the models become more complex, they also become computationally demanding. It has been commonly observed that simple and easily computable measures like common neighbors often have competitive performance with more complex methods.

This behavior has been empirically established across a variety of networks, starting from co-authorship networks [8] to router level internet connections, protein protein interaction networks and electrical power grid network [9].

**Theoretical results:** Spectral clustering has been shown to asymptotically recover cluster memberships for variations of Stochastic Blockmodels [10, 4, 13]. However, apart from [15], there is little understanding of why simple methods such as common neighbors perform so well empirically.

Given their empirical success and computational tractability, the common neighbors heuristic is widely applied for large networks. Understanding the reasons for the accuracy of common neighbors under the popular stochastic blockmodel setting is the goal of our work.

## 3 Proposed Work

Many link prediction methods ultimately make two assumptions: (a) each node belongs to a latent "cluster", where nodes in the same cluster have similar behavior; and (b) each node is very likely to connect to others in its cluster, so link prediction is equivalent to finding other nodes in the cluster. These assumptions can be relaxed: instead of belonging to the same cluster, nodes could have "topic distributions", with links being more likely between pairs of nodes with similar topical interests. However, we will focus on the assumptions stated above, since they are clean and the relaxations appear to be fundamentally similar.

**Model.** Specifically, consider a stochastic blockmodel where each node $i$ belongs to an unknown cluster $c_i \in \{C_1, \ldots, C_K\}$. We assume that the number of clusters $K$ is fixed as the number of nodes $n$ increases. We also assume that each cluster has $\pi = n/K$ members, though this can be relaxed easily. The probability $P(i \sim j)$ of a link between nodes $i$ and $j$ $(i \neq j)$ depends only on the clusters of $i$ and $j$: $P(i \sim j) = B_{c_i, c_j} \triangleq \alpha\{c_i = c_j\} + \gamma\{c_i \neq c_j\}$ for some $\alpha > \gamma > 0$; in other words, the probability of a link is $\alpha$ between nodes in the same cluster, and $\gamma$ otherwise. By definition, $P(i \sim i) = 0$. If the nodes were arranged so that all nodes in a cluster are contiguous, then the corresponding matrix, when plotted, attains a block-like structure, with the diagonal blocks (corresponding to links within a cluster) being denser than off-diagonal blocks (since $\alpha > \gamma$).

Under these assumptions, we ask the following two questions:

**Problem 1** (Link Prediction and Recommendation). *Given node $i$, how can we identify at least a constant number of nodes from $c_i$?*

**Problem 2** (Local Cluster Detection). *Given node $i$, how can we identify all nodes in $c_i$?*

Problem 1 can be considered as the problem of finding good recommendations for a given node $i$. Here, the goal is to find a few good nodes that $i$ could connect to (e.g., recommending a few possible friends on Facebook, or a few movies to watch next on Netflix). Since within-cluster links have higher probability than across-cluster links ($\alpha > \gamma$), predicting nodes from $c_i$ gives the optimal answer. Crucially, it is unnecessary to find *all* good nodes. As against that, Problem 2 requires us to find everyone in the given node's cluster. This is the problem of detecting the entire cluster corresponding to a given node. Note that Problem 2 is clearly harder than Problem 1.

We next present a summary of our results and the underlying intuition before delving into the details.

### 3.1 Intuition and Result Summary

**Current approaches.** Standard approaches to inference for the stochastic blockmodel attempt to solve an even harder problem:

**Problem 3** (Full Cluster Detection). *How can we identify the latent clusters $c_i$ for all $i$?*

A popular solution is via *spectral clustering*, involving two steps: (a) computing the top-$K$ eigenvectors of the graph Laplacian, and (b) clustering the projections of each node on the

corresponding eigenspace via an algorithm like k-means [13]. A slight variation of this has been shown to work as long as $(\alpha - \gamma)/\sqrt{\alpha} = \Omega(\log n/\sqrt{n})$ and the average degree grows faster than poly-logarithmic powers of $n$ [10].

However, (a) spectral clustering solves a harder problem than Problems 1 and 2, and (b) eigen-decompositions can be expensive, particularly for very large graphs. Our claim is that a simpler operation — counting common neighbors between nodes — can yield results that are almost as good in a broad parameter regime.

**Common neighbors.** Given a node $i$, link prediction via common neighbors follows a simple prescription: predict a link to node $j$ such that $i$ and $j$ have the maximum number $|CN(i,j)|$ of shared friends $CN(i,j) = \{u \mid i \sim u \sim j\}$. The usefulness of common neighbors have been observed in practice [8] and justified theoretically for the latent distance model [15]. However, its properties under the stochastic blockmodel remained unknown.

Intuitively, we would expect a pair of nodes $i$ and $j$ from the same cluster to have many common neighbors $u$ from the same cluster, since both the links $i \sim u$ and $u \sim j$ occur with probability $\alpha$, whereas for $c_i \neq c_j$, at least one of the edges $i \sim u$ and $u \sim j$ must have the lower probability $\gamma$.

$$P(u \in CN(i,j) \mid c_i = c_j) = \alpha^2 P(c_u = c_i \mid c_i = c_j) + \gamma^2 P(c_u \neq c_i \mid c_i = c_j)$$
$$= \pi\alpha^2 + (1-\pi)\gamma^2$$
$$P(u \in CN(i,j) \mid c_i \neq c_j) = \alpha\gamma P(c_u = c_i \text{ or } c_u = c_j \mid c_i \neq c_j) + \gamma^2 P(c_u \neq c_i, c_u \neq c_j \mid c_i \neq c_j)$$
$$= 2\pi\alpha\gamma + (1-2\pi)\gamma^2 = P(u \in CN(i,j) \mid c_i = c_j) - \pi(\alpha-\gamma)^2$$
$$\leq P(u \in CN(i,j) \mid c_i = c_j)$$

Thus the expected number of common neighbors $E\left[|CN(i,j)|\right]$ is higher when $c_i = c_j$. If we can show that the random variable $CN(i,j)$ concentrates around its expectation, node pairs with the most common neighbors would belong to the same cluster. Thus, common neighbors would offer a good solution to Problem 1.

We show conditions under which this is indeed the case. There are three key points regarding our method: (a) handling dependencies between common neighbor counts, (b) defining the graph density regime under which common neighbors is consistent, and (c) proposing a variant of common neighbors which significantly broadens this region of consistency.

**Dependence.** $CN(i,j)$ and $CN(i,j')$ are dependent; hence, distinguishing between within-group and outside-group nodes can be complicated even if each $CN(i,j)$ concentrates around its expectation. We handle this via a careful conditioning step.

**Dense versus sparse graphs.** In general, the parameters $\alpha$ and $\gamma$ can be functions of $n$, and we can try to characterize parameter settings when common neighbors consistently returns nodes from the same cluster as the input node. We show that when the graph is sufficiently "dense" (average degree is growing faster than $\sqrt{n \log n}$), common neighbors is powerful enough to answer Problem 2. Also, $(\alpha - \gamma)/\alpha$ can go to zero at a suitable rate.

On the other hand, the expected number of common neighbors between nodes tends to zero for sparser graphs, irrespective of whether the nodes are in the same cluster or not. Further, the standard deviation is of a higher order than the expectation, so there is no concentration. In this case, counting common neighbors fails, even for Problem 1.

**A variant with better consistency properties.** However, we show that the addition of an extra post-processing step (henceforth, the "cleaning" step) still enables common neighbors to identify nodes from its own cluster, while reducing the number of off-cluster nodes to zero with probability tending to one as $n \to \infty$. This requires a stronger separation condition between $\alpha$ and $\gamma$. However, such "strong consistency" is only possible when the average degree grows faster than $(n \log n)^{1/3}$. Thus, the cleaning step extends the consistency of common neighbors beyond the $O(1/\sqrt{n})$ range.

## 4 Main Results

We first split the edge set of the complete graph on $n$ nodes into two sets: $K_1$ and its complement $K_2$ (independent of the given graph $G$). We compute common neighbors on $G_1 = G \cap K_1$ and perform a "cleaning" process on $G_2 = G \cap K_2$. The adjacency matrices of $G_1$ and $G_2$ are denoted by $A_1$ and $A_2$. We will fix a reference node $q$, which belongs to class $C_1$ without loss of generality (recall that there are $K$ clusters $C_1 \ldots C_K$, each of size $n\pi$).

Let $X_i (i \neq q)$ denote the number of common neighbors between $q$ and $i$. Algorithm 1 computes the set $S = \{i : X_i \geq t_n\}$ of nodes who have at least $t_n$ common neighbors with $q$ on $A_1$, whereas Algorithm 2 does a further degree thresholding on $A_2$ to refine $S$ into $S_1$.

---
**Algorithm 1** Common neighbors screening algorithm
---
1: **procedure** $\textsc{Scan}(A_1, q, t_n)$
2:     For $1 \leq i \leq n$, $X_i \leftarrow A_1^2(q, i)$
3:     $X_q \leftarrow 0$
4:     $S \leftarrow \{i : X_i \geq t_n\}$
5:     **return** $S$
---

---
**Algorithm 2** Post Selection Cleaning algorithm
---
1: **procedure** $\textsc{Clean}(S, A_2, q, s_n)$
2:     $S_1 \leftarrow \{i : \sum_{j \in S} A_2(i, j) \geq s_n\}$
3:     **return** $S_1$
---

To analyze the algorithms, we must specify conditions on graph densities. Recall that $\alpha$ and $\gamma$ represent within-cluster and across-cluster link probabilities. We assume that $\alpha/\gamma$ is constant while $\alpha \to 0, \gamma \to 0$; equivalently, assume that both $\alpha$ and $\gamma$ are both some constant times $\rho$, where $\rho \to 0$.

The analysis of graphs has typically been divided into two regimes. The *dense* regime consists of graphs with $n\rho \to \infty$, where the expected degree $n\rho$ is a fraction of $n$ as $n$ grows. In the *sparse* regime, $n\rho = O(1)$, so degree is roughly constant. Our work explores a finer gradation, which we call *semi-dense* and *semi-sparse*, defined next.

**Definition 4.1** (Semi-dense graph)**.** *A sequence of graphs is called semi-dense if $n\rho^2 / \log n \to \infty$ as $n \to \infty$.*

**Definition 4.2** (Semi-sparse graph)**.** *A sequence of graphs is called semi-sparse if $n\rho^2 \to 0$ but $n^{2/3}\rho / \log n \to \infty$ as $n \to \infty$.*

Our first result is that common neighbors is enough to solve not only the link-prediction problem (Problem 1) but also the local clustering problem (Problem 2) in the semi-dense case. This is because even though both nodes within and outside the query node's cluster have a growing number of common neighbors with $q$, there is a clear distinction in the expected number of common neighbors between the two classes. Also, since the standard deviation is of a smaller order than the expectation, the random variables concentrate. Thus, we can pick a threshold $t_n$ such that $\textsc{Scan}(A_1, q, t_n)$ yields just the nodes in the same cluster as $q$ with high probability. Note that the cleaning step (Algorithm 2) is not necessary in this case.

**Theorem 4.1** (Algorithm 1 solves Problem 2 in semi-dense graphs)**.** *Let $t_n = n \left( \pi(\alpha + \gamma)^2 / 2 + (1 - 2\pi)\gamma^2 \right)$. Let $S$ be the set of nodes returned by $\textsc{Scan}(A_1, q, t_n)$. Let $n_w$ and $n_o$ denote the number of nodes in $S \cap C_1$ and $S \setminus C_1$ respectively. If the graph is semi-dense, and if $\frac{\alpha - \gamma}{\alpha} \geq \frac{2}{\sqrt{\pi}} \left( \frac{\log n}{n\alpha^2} \right)^{1/4}$, then $P(n_w = n\pi) \to 1$ and $P(n_o = 0) \to 1$.*

*Proof Sketch.* We only sketch the proof here, deferring details to the supplementary material. Let $d_{qa} = \sum_{i \in C_a} A_1(q, i)$ be the number of links from the query node $q$ to nodes in

cluster $C_a$. Let $\mathbf{d}_q = \{d_{q1}, \dots q_{qK}\}$ and $d = \sum_a d_{qa}$. We first show that

$$P(\mathbf{d}_q \in \text{GOOD}) \triangleq P\left(\begin{array}{cc} d_{q1} \in n\pi\alpha(1 \pm \psi_n) \\ d_{qa} \in n\pi\gamma(1 \pm \psi_n) & \forall a \neq 1 \end{array}\right) \geq 1 - \frac{K}{n^2}, \tag{1}$$

$$\psi_n \triangleq \sqrt{(6\log n)/(n\pi\gamma)} = \sqrt{\sqrt{\log n/n} \cdot \Theta(\sqrt{\log n/(n\rho^2)})} \to 0. \tag{2}$$

Conditioned on $\mathbf{d}_q$, $X_i$ is the sum of $K$ Binomial$(d_{qa}, B_{1a})$ independent random variables representing the number of common neighbors between $q$ and $i$ via nodes in each of the $K$ clusters: $E[X_i \mid \mathbf{d}_q, i \in C_a] = d_{qa}\alpha + (d - d_{qa})\gamma$. We have:

$$\eta_1 \triangleq E[X_i \mid \mathbf{d}_q \in \text{GOOD}, i \in C_1] \geq n\left(\pi\alpha^2 + (1-\pi)\gamma^2\right)(1 - \psi_n) \triangleq \ell_n(1 - \psi_n)$$

$$\eta_a \triangleq E[X_i \mid \mathbf{d}_q \in \text{GOOD}, i \in C_a, a \neq 1] \leq n\left(2\pi\alpha\gamma + (1-2\pi)\gamma^2\right)(1 + \psi_n) \triangleq u_n(1 + \psi_n)$$

Note that $t_n = (\ell_n + u_n)/2$, $u_n \leq t_n \leq \ell_n$, and $\ell_n - u_n = n\pi(\alpha - \gamma)^2 \geq 4\log n\sqrt{n\alpha^2/\log n} \to \infty$, where we applied condition on $(\alpha - \gamma)/\alpha$ noted in the theorem statement. We show:

$$P\left(X_i \leq t_n \mid \mathbf{d}_q \in \text{GOOD}, i \in C_1\right) \leq n^{-4/3 + o(1)}$$

$$P\left(X_i \geq t_n \mid \mathbf{d}_q \in \text{GOOD}, i \in C_a, a \neq 1\right) \leq n^{-4/3 + o(1)}$$

Conditioned on $\mathbf{d}_q$, both $n_w$ and $n_o$ are sums of conditionally independent and identically distributed Bernoullis.

$$P(n_w = n\pi) \geq P(\mathbf{d}_q \in \text{GOOD})P(n_w = n\pi \mid \mathbf{d}_q \in \text{GOOD}) \geq \left(1 - \frac{K}{n^2}\right) \cdot (1 - n\pi \cdot n^{-4/3}) \to 1$$

$$P(n_o = 0) \geq P(\mathbf{d}_q \in \text{GOOD}) \cdot P(n_o = 0 \mid \mathbf{d}_q \in \text{GOOD}) \geq 1 - \Theta(n^{-1/3}) \to 1$$

$\square$

There are two major differences between the semi-sparse and semi-dense cases. First, in the semi-sparse case, both expectations $\eta_1$ and $\eta_a$ are of the order $O(n\rho^2)$ which tends to zero. Second, standard deviations on the number of common neighbors are of a larger order than expectations. Together, this means that the number of common neighbors to within-cluster and outside-cluster nodes can no longer be separated; hence, Algorithm 1 by itself cannot work. However, *after* cleaning, the entire cluster of the query node $q$ can still be recovered.

**Theorem 4.2** (Algorithm 1 followed by Algorithm 2 solves Problem 2 in semi-sparse graphs). *Let $t_n = 1$ and $s_n = n^2(\pi\alpha + (1-\pi)\gamma)^2(\alpha + \gamma)/2$. Let $S = \text{SCAN}(A_1, q, t_n)$ and $S_1 = \text{CLEAN}(S, A_2, q, s_n)$. Let $n_w^{(c)}\left(n_o^{(c)}\right)$ denote the number of nodes in $S_1 \cap C_1$ $(S_1 \setminus C_1)$. If the graph is semi-sparse, and $\pi\alpha \geq 3(1-\pi)\gamma$, then $P\left(n_w^{(c)} = n\pi\right) \to 1$ and $P\left(n_o^{(c)} = 0\right) \to 1$.*

*Proof Sketch.* We only sketch the proof here, with details being deferred to the supplementary material. The degree bounds of Eq. 1 and the equations for $E[X_i | \mathbf{d}_q \in \text{GOOD}]$ hold even in the semi-sparse case. We can also bound the variances of $X_i$ (which are sums of conditionally independent Bernoullis):

$$\text{var}[X_i \mid \mathbf{d}_q \in \text{GOOD}, i \in C_1] \leq E[X_i \mid \mathbf{d}_q \in \text{GOOD}, i \in C_1] = \eta_1$$

Since the expected number of common neighbors vanishes and the standard deviation is an order larger than the expectation, there is no hope for concentration; however, there are slight differences in the probability of having at least one common neighbor.

First, by an application of the Paley-Zygmund inequality, we find:

$$p_1 \triangleq P(X_i \geq 1 \mid \mathbf{d}_q \in \text{GOOD}, i \in C_1)$$

$$\geq \frac{E[X_i \mid \mathbf{d}_q \in \text{GOOD}, i \in C_1]^2}{\text{var}(X_i \mid \mathbf{d}_q \in \text{GOOD}, i \in C_1) + E[X_i \mid \mathbf{d}_q \in \text{GOOD}, i \in C_1]^2}$$

$$\geq \frac{\eta_1^2}{\eta_1 + \eta_1^2} \geq \ell_n(1 - \psi_n)(1 - \eta_1) \qquad \text{since } \eta_1 \to 0$$

For $a > 1$, Markov's inequality gives:

$$p_a \triangleq P(X_i \geq 1 \mid \mathbf{d}_q \in \textsc{Good}, i \in C_a, a \neq 1) \leq E[X_i \mid \mathbf{d}_q \in \textsc{Good}, i \in C_a, a \neq 1] = \eta_a$$

Even though $p_a \to 0$, $n\pi p_a = \Theta(n^2\rho^2) \to \infty$, so we can use concentration inequalities like the Chernoff bound again to bound $n_w$ and $n_o$.

$$P(n_w \geq n\pi p_1(1 - \sqrt{6\log n/n\pi p_1})) \geq 1 - n^{-4/3}$$
$$P(n_o \leq n(1-\pi)p_a(1 + \sqrt{6\log n/n(1-\pi)p_a})) \geq 1 - n^{-4/3}$$

Unlike the denser regime, $n_w$ and $n_o$ can be of the same order here. Hence, the candidate set $S$ returned by thresholding the common neighbors has a non-vanishing fraction of nodes from outside $q$'s community. However, this fraction is relatively small, which is what we would exploit in the cleaning step.

Let $\theta_w$ and $\theta_o$ denote the expected number of edges in $A_2$ from a node to $S$. The separation condition in the theorem statement gives $\theta_w - \theta_o \geq 4\sqrt{\theta_w \log n}$. Setting the degree threshold $s_n = (\theta_w + \theta_o)/2$, we bound the probability of mistakes in the cleaning step:

$$P(\exists i \in C_1 \text{ s.t. } \sum_{j \in S} A_2(i,j) \leq s_n \mid \mathbf{d}_q \in \textsc{Good}) \leq n^{-1/3+o(1)}$$
$$P(\exists i \notin C_1 \text{ s.t. } \sum_{j \in S} A_2(i,j) \geq s_n \mid \mathbf{d}_q \in \textsc{Good}) \leq n^{-1/3+o(1)}$$

Removing the conditioning on $\mathbf{d}_q \in \textsc{Good}$ (as in Theorem 4.1) yields the desired result. $\square$

## 5  Experiments

We present our experimental results in two parts. First, we use simulations to support our theoretical claims. Next we present link prediction accuracies on real world collaborative networks to show that common neighbors indeed perform close to gold standard algorithms like spectral clustering and the Katz score.

**Implementation details:** Recall that our algorithms are based on thresholding. When there is a large gap between common neighbors between node $q$ and nodes in its cluster (e.g., in the semi-dense regime), this is equivalent to using the k-means algorithm with $k = 2$ to find $S$ in Algorithm 1. The same holds for finding $S_1$ in algorithm 2. When the number of nodes with more than two common neighbors is less than ten, we define the set $S$ by finding all neighbors with at least one common neighbor (as in the semi-sparse regime). On the other hand, since the cleaning step works only when $S$ is sufficiently large (so that degrees concentrate), we do not perform any cleaning when $|S| < 30$. While we used the split sample graph $A_2$ in the cleaning step for ease of analysis, we did the cleaning using the same network in the experiments.

**Experimental setup for simulations:** We use a stochastic blockmodel of 2000 nodes split into 4 equal-sized clusters. For each value of $(\alpha, \gamma)$ we pick 50 query nodes at random, and calculate the precision and recall of the result against nodes from the query node's cluster (for any subset $S$ and true cluster $C$, precision $= |S \cap C|/|S|$ and recall $= |S \cap C|/|C|$). We report mean precision and recall over 50 random generated graph instances.

**Accuracy on simulated data:** Figure 1 shows the precision and recall as degree grows, with the parameters $(\alpha, \gamma)$ satisfying the condition $\pi\alpha \geq 3(1-\pi)\gamma$ of Thm. 4.2. We see that cleaning helps both precision and recall, particularly in the medium-degree range (the semi-sparse regime). As a reference, we also plot the precision of spectral clustering, when it was given the correct number of clusters ($K = 4$). Above average degree of 10, spectral clustering gives perfect precision, whereas common neighbors can identify a large fraction of the true cluster once average degree is above 25. On the other hand, for average degree less than seven, spectral clustering performs poorly, whereas the precision of common neighbors is remarkably higher. Precision is relatively higher than recall for a broad degree regime, and this explains why common neighbors are a popular choice for link prediction. On a side

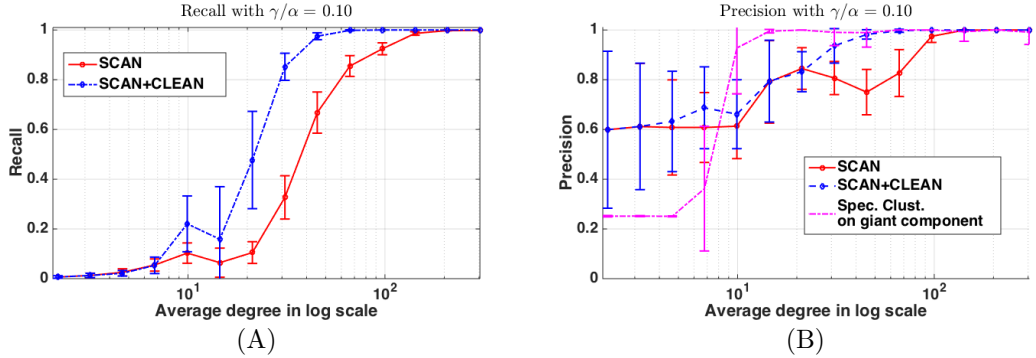

Figure 1: *Recall and precision versus average degree:* When degree is very small, none of the methods work well. In the medium-degree range (semi-sparse regime), we see that common neighbors gets increasingly better precision and recall, and cleaning helps. With high enough degrees (semi-dense regime), just common neighbors is sufficient and gets excellent accuracy.

Table 1: AUC scores for co-authorship networks

| Dataset | n | Mean degree | Time-steps | AUC | | | | |
|---------|------|-------------|------------|-----|----------|------|------|--------|
|         |      |             |            | CN  | CN-clean | SPEC | Katz | Random |
| HepTH   | 5969 | 4           | 6          | .70 | .74      | .82  | .82  | .49    |
| Citeseer| 4520 | 5           | 11         | .88 | .89      | .89  | .95  | .52    |
| NIPS    | 1222 | 3.95        | 9          | .63 | .69      | .68  | .78  | .47    |

note, it is not surprising that in a very sparse graph common neighbors cannot identify the whole cluster, since not everyone can be reached in two hops.

**Accuracy on real-world data:** We used publicly available co-authorship datasets over time where nodes represent authors and an edge represents a collaboration between two authors. In particular, we used subgraphs of the High Energy Physics (HepTH) co-authorship dataset (6 timesteps), the NIPS dataset (9 timesteps) and the Citeseer dataset (11 timesteps). We obtain the training graph by merging the first T-2 networks, use the T-1$^{th}$ step for cross-validation and use the last timestep as the test graph. The number of nodes and average degrees are reported in Table 1. We merged 1-2 years of papers to create one timestep (so that the median degree of the test graph is at least 1).

We compare our algorithm (CN and CN-clean) with the Katz score which is used widely in link prediction [8] and spectral clustering of the network. Spectral clustering is carried out on the giant component of the network. Furthermore, we cross-validate the number of clusters using the held out graph. Our setup is very similar to link prediction experiments in related literature [14].

Since these datasets are unlabeled, we cannot calculate precision or recall as before. Instead for any score or affinity measure, we propose to perform link prediction experiments as follows. For a randomly picked node we calculate the score from the node to everyone else. We compute the AUC score of this vector against the edges in the test graph. We report the average AUC for 100 randomly picked nodes. Table 1 shows that even in sparse regimes common neighbors performs similar to benchmark algorithms.

## 6   Conclusions

Counting common neighbors is a particularly useful heuristic: it is fast and also works well empirically. We prove the effectiveness of common neighbors for link prediction as well as local clustering around a query node, under the stochastic blockmodel setting. In particular, we show the existence of a semi-dense regime where common neighbors yields the right cluster w.h.p, and a semi-sparse regime where an additional "cleaning" step is required. Experiments with simulated as well as real-world datasets shows the efficacy of our approach, including the importance of the cleaning step.

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
