[Supplementary Material]

# Supplementary Material

**Purnamrita Sarkar**
Department of Statistics
University of Texas at Austin
purnamritas@austin.utexas.edu

**Deepayan Chakrabarti**
IROM, McCombs School of Business
University of Texas at Austin
deepay@utexas.edu

**Peter Bickel**
Department of Statistics
University of California, Berkeley
bickel@stat.berkeley.edu

**Theorem 4.1** (Algorithm 1 solves Problem 2 in semi-dense graphs). *Let* $t_n = n\left(\pi(\alpha+\gamma)^2/2 + (1-2\pi)\gamma^2\right)$. *Let* $S$ *be the set of nodes returned by* $\mathrm{SCAN}(A_1, q, t_n)$. *Let* $n_w$ *($n_o$) denote the number of nodes in* $S \cap C_1$ *($S \setminus C_1$). If the graph is semi-dense, and* $\frac{\alpha-\gamma}{\alpha} \geq \frac{2}{\sqrt{\pi}}\left(\frac{\log n}{n\alpha^2}\right)^{1/4}$, *then* $P(n_w = n\pi) \to 1$ *and* $P(n_o = 0) \to 1$.

*Proof.* Let $d_{qa} = \sum_{i \in C_a} A_1(q, i)$ be the number of links from the query node $q$ to nodes in cluster $C_a$. Let $\mathbf{d}_q = \{d_{q1}, \ldots q_{qK}\}$ and $d = \sum_a d_{qa}$.

Let $\psi_n = \sqrt{(6 \log n)/(n\pi\gamma)}$. By a Chernoff bound, we can show that

$$P\left(|d_{q1} - n\pi\alpha| \leq n\pi\alpha\psi_n\right) \leq 2/n^2 \tag{1}$$

$$P\left(|d_{qa} - n\pi\gamma| \leq n\pi\gamma\psi_n\right) \leq 2/n^2 \quad \forall a \neq 1 \tag{2}$$

$$\Rightarrow P(\mathbf{d}_q \in \textsc{Good}) \triangleq P\left(\begin{array}{l} d_{q1} \in n\pi\alpha(1 \pm \psi_n) \\ d_{qa} \in n\pi\gamma(1 \pm \psi_n) \quad \forall a \neq 1 \end{array}\right) \geq 1 - \frac{K}{n^2}, \tag{3}$$

where the $\textsc{Good}$ set is defined via the last inequality. Note that

$$\psi_n = \sqrt{\Theta\left(\log n/(n\rho)\right)} = \sqrt{\sqrt{\log n/n} \cdot \Theta(\sqrt{\log n/(n\rho^2)})} \to 0. \tag{4}$$

Conditioned on $\mathbf{d}_q$, $X_i$ is the sum of $K$ Binomial$(d_{qa}, B_{1a})$ independent random variables representing the number of common neighbors between $q$ and $i$ via nodes in each of the $K$ clusters:

$$\hat{\eta}_a \triangleq E[X_i \mid \mathbf{d}_q, i \in C_a] = d_{qa}\alpha + (d - d_{qa})\gamma.$$

We have, for $\mathbf{d}_q \in \textsc{Good}$:

$$n\left(\pi\alpha^2 + (1-\pi)\gamma^2\right)(1 - \psi_n) \leq \hat{\eta}_1 \leq n\left(\pi\alpha^2 + (1-\pi)\gamma^2\right)(1 + \psi_n) \tag{5}$$

$$n\left(2\pi\alpha\gamma + (1-2\pi)\gamma^2\right)(1 - \psi_n) \leq \hat{\eta}_a \leq n\left(2\pi\alpha\gamma + (1-2\pi)\gamma^2\right)(1 + \psi_n) \quad \text{For } a \neq 1 \tag{6}$$

Let us denote by $\ell_n \triangleq n\left(\pi\alpha^2 + (1-\pi)\gamma^2\right)$ and $u_n \triangleq \left(2\pi\alpha\gamma + (1-2\pi)\gamma^2\right)$, and also let $t_n = (\ell_n + u_n)/2$. Clearly, $u_n \leq t_n \leq \ell_n$, and $\ell_n - u_n = n\pi(\alpha-\gamma)^2 \geq 4\log n\sqrt{n\alpha^2/\log n} \to \infty$, where we applied condition on $(\alpha - \gamma)/\alpha$ noted in the theorem statement. Second, we can easily see that $\hat{\eta}_a \leq \hat{\eta}_1 \leq n\alpha^2(1 + \psi_n)$ for large enough $n$.

Now, by a Chernoff bound,

$$P\left(X_i \leq t_n \mid \mathbf{d}_q \in \text{Good}, i \in C_1\right) = E[P\left(X_i \leq t_n \mid \mathbf{d}_q, \mathbf{d}_q \in \text{Good}, i \in C_1\right) \mid \mathbf{d}_q \in \text{Good}]$$

$$\leq E\left[\exp\left(-\frac{(\hat{\eta}_1 - t_n)^2}{3\hat{\eta}_1}\right) \mid \mathbf{d}_q \in \text{Good}\right]$$

$$\leq \exp\left(-\frac{(\ell_n - t_n - \ell_n\psi_n)^2}{3n\alpha^2(1+\psi_n)}\right)$$

$$= \exp\left(-\frac{\left(\frac{\ell_n - u_n}{2}\right)^2\left(1 - \frac{2\ell_n\psi_n}{\ell_n - u_n}\right)^2}{3n\alpha^2(1+\psi_n)}\right)$$

$$\leq \exp\left(-\frac{(\ell_n - u_n)^2}{12n\alpha^2}\left(1 - O\left(\frac{\ell_n\psi_n}{\ell_n - u_n}\right) - O\left(\psi_n\right)\right)\right)$$

(7)

Now,

$$\frac{(\ell_n - u_n)^2}{12n\alpha^2} = \frac{n^2\pi^2(\alpha - \gamma)^4}{12n\alpha^2} \geq 4/3 \log n$$

$$\frac{\ell_n\psi_n}{\ell_n - u_n} = \Theta\left(\frac{n\rho^2}{n\pi(\alpha - \gamma)^2}\sqrt{\frac{\log n}{n\rho}}\right) \leq \Theta\left(\frac{n\rho^2}{\sqrt{n\rho^2 \log n}}\sqrt{\frac{\log n}{n\rho}}\right) = \Theta\left(\sqrt{\rho}\right) \to 0$$

$$\psi_n \to 0, \quad (Eq.\ 2)$$

where we used the condition on $(\alpha - \gamma)/\alpha$, and the fact that $\alpha = \Theta(\rho)$ and $\gamma = \Theta(\rho)$. Using this in Eq. 7 yields

$$P\left(X_i \leq t_n \mid \mathbf{d}_q \in \text{Good}, i \in C_1\right) \leq n^{-4/3+o(1)}.$$

By a similar argument, we find that

$$P\left(X_i \geq t_n \mid \mathbf{d}_q \in \text{Good}, i \in C_a, a \neq 1\right) \leq n^{-4/3+o(1)}.$$

We want to point out that while it seems that we need $\rho \to 0$ for our analysis, that is not the case. In order to analyze the case where $\rho = \Theta(1)$, we would simply need an updated separation condition:

$$(\ell_n - u_n) \geq \max(4\sqrt{n\alpha^2 \log n}, C\psi_n\ell_n)$$

.

When $\rho \to 0$, the first term is larger. This again requires an updated separation between $\alpha$ and $\gamma$, namely

$$(\alpha - \gamma)/\alpha \geq \max\left(\frac{2}{\sqrt{\pi}}\left(\frac{\log n}{n\alpha^2}\right)^{1/4}, \frac{C}{\pi^{3/4}}\left(\frac{\log n}{n}\right)^{1/4}\right),$$

for some large enough constant $C$. However for ease of exposition we only present the $\rho \to 0$ case in the main paper.

Let $Y_i := \mathbf{1}\{X_i \geq t_n\}$. $\text{SCAN}(A_1, q, t_n)$ returns exactly the nodes $S = \{i \mid Y_i = 1\}$. We have:

$$n_w = \sum_{i \in C_1} Y_i \qquad n_o = \sum_{i \notin C_1} Y_i$$

(8)

Conditioned on $\mathbf{d}_q$, both $n_w$ and $n_o$ are sums of conditionally independent and identically distributed Bernoullis.

$$
\begin{aligned}
P(n_w = n\pi) &\geq P(\mathbf{d}_q \in \textsc{Good}) \cdot P(n_w = n\pi \mid \mathbf{d}_q \in \textsc{Good}) \\
&\geq P(\mathbf{d}_q \in \textsc{Good}) \cdot (1 - P(\exists i \in C_1, X_i < t_n | \mathbf{d}_q \in \textsc{Good})) \\
&\geq \left(1 - \frac{K}{n^2}\right) \cdot (1 - n\pi \cdot n^{-4/3}) \\
&\geq 1 - \Theta(n^{-1/3}) \\
&\to 1 \\
P(n_o = 0) &\geq P(\mathbf{d}_q \in \textsc{Good}) \cdot P(n_o = 0 \mid \mathbf{d}_q \in \textsc{Good}) \\
&\geq P(\mathbf{d}_q \in \textsc{Good}) \cdot (1 - P(\exists i \notin C_1, X_i \geq t_n | \mathbf{d}_q \in \textsc{Good})) \\
&\geq \left(1 - \frac{K}{n^2}\right) \cdot (1 - n(1 - \pi) \cdot n^{-4/3}) \\
&\geq 1 - \Theta(n^{-1/3}) \\
&\to 1
\end{aligned}
$$

$\square$

**Theorem 4.2** (Algorithm 1 followed by Algorithm 2 solves Problem 2 in semi-sparse graphs). *Let $t_n = 1$ and $s_n = n^2 \left(\pi\alpha + (1 - \pi)\gamma\right)^2 (\alpha + \gamma)/2$. Let $S = \textsc{Scan}(A_1, q, t_n)$ and $S_1 = \textsc{Clean}(S, A_2, q, s_n)$. Let $n_w^{(c)} \left(n_o^{(c)}\right)$ denote the number of nodes in $S_1 \cap C_1$ $(S_1 \setminus C_1)$. If the graph is semi-sparse, and $\pi\alpha \geq 3(1 - \pi)\gamma$, then $P\left(n_w^{(c)} = n\pi\right) \to 1$ and $P\left(n_o^{(c)} = 0\right) \to 1$.*

*Proof.* The degrees of nodes can still be bound w.h.p. via Eq. 1 since in the semi-sparse case

$$
\psi_n = \sqrt{\Theta\left(\log n/(n\rho)\right)} = \sqrt{\frac{1}{n^{1/3}} \cdot \frac{\log n}{n^{2/3}\rho}} \to 0.
$$

Similarly, the equations for the $E[X_i \mid \mathbf{d}_q \in \textsc{Good}]$ hold as well (Eqs. 5 and 6). We can also bound the variances of $X_i$ (which are sums of conditionally independent Bernoullis):

$$
\begin{aligned}
\mathrm{var}[X_i \mid \mathbf{d}_q, i \in C_1] &= d_{q1}\alpha(1 - \alpha) + (d - d_{q1})\gamma(1 - \gamma) \\
&\leq E[X_i \mid \mathbf{d}_q, i \in C_1] \triangleq \hat{\eta}_1 \qquad\qquad \text{Since } \gamma < \alpha < 1
\end{aligned}
$$

These highlight two major differences between the semi-sparse and semi-dense cases. First, in the semi-sparse case, both expectations $\hat{\eta}_1$ and $\hat{\eta}_a$ (for $\mathbf{d}_q \in \textsc{Good}$) are of the order $O(n\rho^2)$ which tends to zero. Second, standard deviations on the number of common neighbors are of a larger order than expectations. Together, this means that the number of common neighbors to within-cluster and outside-cluster nodes can no longer be separated; hence, Algorithm 1 by itself cannot work.

In spite of this, there are small differences between nodes within and outside the query cluster, which can be exploited. First, by an application of the Paley-Zygmund inequality, we find a lower bound as:

$$
\begin{aligned}
p_a &\triangleq P(X_i \geq 1 \mid \mathbf{d}_q, i \in C_a) \\
&\geq \frac{E[X_i \mid \mathbf{d}_q, i \in C_a]^2}{\mathrm{var}(X_i \mid \mathbf{d}_q, i \in C_a) + E[X_i \mid \mathbf{d}_q, i \in C_a]^2} \\
&\geq \frac{\hat{\eta}_a^2}{\hat{\eta}_a + \hat{\eta}_a^2} \geq \hat{\eta}_a(1 - \hat{\eta}_a)
\end{aligned}
$$

On the other hand Markov's inequality can be used to upper bound this quantity:

$$
p_a \leq E(X_i \mid \mathbf{d}_q, i \in C_a) = \hat{\eta}_a
$$

Hence for $a = 1$ vs $a \neq 1$, using Equations 5 and 6 we have:

For $\mathbf{d}_q \in \textsc{Good}$
$$\ell_n(1 - \xi_n) \leq \hat{\eta}_1(1 - \hat{\eta}_1) \leq p_1 \leq \hat{\eta}_1 \leq \ell_n(1 + \psi_n) \tag{9}$$
$$u_n(1 - \xi'_n) \leq \hat{\eta}_a(1 - \hat{\eta}_a) \leq p_a \leq \hat{\eta}_a \leq u_n(1 + \psi_n) \tag{10}$$

where $\xi_n \triangleq \psi_n + \ell_n + 2\psi_n\ell_n + \ell_n\psi_n^2$ and $\xi'_n \triangleq \psi_n + u_n + 2\psi_n u_n + u_n\psi_n^2$.

Note that even though $p_a \to 0$ in probability, w.h.p. (when $\mathbf{d}_q \in \textsc{Good}$) $n\pi p_a \to \infty$ faster than $\log n$. So we can use concentration inequalities like the Chernoff bound again to bound $n_w$ and $n_o$.

$$P\left(n\pi p_1\left(1 - \phi_n\right) \leq n_w \leq n\pi p_1\left(1 + \phi_n\right) \mid \mathbf{d}_q\right) \geq 1 - 2n^{-2} \tag{11}$$

Similarly,

$$P(n(1 - \pi)p_a(1 - \delta_n) \leq n_o \leq n(1 - \pi)p_a(1 + \delta_n) \mid \mathbf{d}_q) \geq 1 - 2n^{-2} \tag{12}$$
$$\tag{13}$$

Since $\delta_n \triangleq \sqrt{6\log n/n(1 - \pi)p_a}$ and $\phi_n \triangleq \sqrt{6\log n/n\pi p_1}$ are $O(\sqrt{\log n/n^2\rho^2})$, they are $o(1)$ for $\mathbf{d}_q \in \textsc{Good}$.

Note that unlike the denser regime, $n_w$ and $n_o$ can be of the same order here. And so the candidate set $S$ returned by thresholding the common neighbors has a non-vanishing fraction of nodes from outside $q$'s community. However, this fraction is relatively small, which is what we would exploit in the cleaning step.

We will heavily use the fact that $A_2$ is an independent copy of $A$ and so the number of edges to the set $S$ obtained by thresholding common neighbors from $A$, are still pairwise independent. The expectation of the number of edges from a node to $S$ is given by:

$$\theta_w \triangleq E[\sum_{j \in S} A_2(i, j) \mid i \in C_1, \mathbf{d}_q] = n_w\alpha + n_o\gamma \tag{14}$$

$$\theta_o \triangleq E[\sum_{j \in S} A_2(i, j) \mid i \notin C_1, \mathbf{d}_q] = n_w\gamma + n_o\alpha \tag{15}$$

Now we will bound the probability of mistakes in the cleaning step. We set the degree threshold $s_n = n(\alpha + \gamma)(\pi\alpha + (1 - \pi)\gamma)^2/2$.

Using Equations 10 we have with probability at least $1 - 2/n^2$,

$$\theta_w = n_w\alpha + n_o\gamma \geq n\pi p_1\alpha(1 - \phi_n) + n(1 - \pi)p_a\gamma(1 - \delta_n)$$
$$\geq n\pi\ell_n\alpha(1 - \xi_n)(1 - \phi_n) + n(1 - \pi)u_n\gamma(1 - \xi'_n)(1 - \delta_n)$$
$$= (n\pi\ell_n\alpha + n(1 - \pi)u_n\gamma) + w_n$$

where $w_n$ is the remainder term, whose magnitude is $o(n^2\rho^3)$, since $\xi_n, \phi_n = o(1)$, when $\mathbf{d}_q \in \textsc{Good}$. Thus we have:

For $\mathbf{d}_q \in \textsc{Good}$
$$\theta_w - s_n \geq (n\pi\ell_n\alpha + n(1 - \pi)u_n\gamma) + w_n - s_n$$
$$\geq (n\pi\ell_n - n(1 - \pi)u_n)(\alpha - \gamma)/2 + w_n$$
$$= n^2(\alpha - \gamma)((\pi\alpha + (1 - \pi)\gamma)(\pi\alpha - (1 - \pi)\gamma) - 2\pi(1 - \pi)\gamma(\alpha - \gamma))/2 + w_n$$
$$= n^2(\alpha - \gamma)(1 - \pi)\gamma^2 + w_n = n^2(\alpha - \gamma)(1 - \pi)\gamma^2(1 + o(1))$$
$$\geq 4\sqrt{\theta_w\log n} \qquad \text{For large enough } n$$

The last step uses the definition of $\ell_n$ and $u_n$, the separation condition between $\alpha$ and $\gamma$ and an algebraic simplification. We also use the fact that for $\mathbf{d}_q \in \textsc{Good}$, $\theta_w \leq n\pi p_1(1 + o(1)) \leq n^2\pi\alpha^3(1 + o(1))$ w.h.p.

A similar argument holds for $s_n - \theta_o$ as well; in fact, $s_n$ was chosen to be the midpoint of the lower bound of $\theta_w$ and the upper bound of $\theta_o$.

Now, the probability of a node from $C_1$ having number of edges to $S$ below the threshold is given by:

$$P(\exists i \in C_1, \sum_{j \in S} A_2(i,j) \leq s_n \mid \mathbf{d}_q \in \textsc{Good}) \leq nP(\sum_{j \in S} A_2(i,j) \geq s_n; i \in C_1 \mid \mathbf{d}_q \in \textsc{Good})$$

$$\leq nE\left[ P\left( \sum_{j \in S} A_2(i,j) \geq s_n; i \in C_1 \mid \mathbf{d}_q \right) \mid \mathbf{d}_q \in \textsc{Good} \right] + c/n^2$$

$$\leq nE\left[ \exp(-(\theta_w - s_n)^2/3\theta_w) \mid \mathbf{d}_q \in \textsc{Good} \right] + c/n^2 \leq n^{-1/3} + cn^{-2} \to 0$$

The $c/n^2$ term comes from the error probabilities in Equations 11 and 12. Similarly the probability of a node from outside $C_1$ having number of edges to $S$ above the threshold $s_n$ can be upper bounded by:

$$P(\exists i \notin C_1, \sum_{j \in S} A_2(i,j) \geq s_n \mid \mathbf{d}_q \in \textsc{Good}) \leq nP(\sum_{j \in S} A_2(i,j) \geq s_n; i \notin C_1 \mid \mathbf{d}_q \in \textsc{Good})$$

$$\leq nE[\exp(-(\theta_o - s_n)^2/3\theta_o) \mid \mathbf{d}_q \in \textsc{Good}] + c/n^2 \leq n^{-1/3} \to 0$$

These two error probabilities and argument identical to the proof of theorem 4.1 establish that $P(S_1 = C_1) \to 0$ under semi-sparse regime.

$\square$