[Reviews · NeurIPS 2015]

Submitted by Assigned_Reviewer_1

This work is about predicting links in graphs using a heuristic based on common neighbours. The aim is to achieve a fast procedure based on counting common neighbours (which can easily be achieved by many informatics systems).

Although just a heuristic method, the performance on real-world data is not far behind benchmark algorithms.

The main manuscript is not does not quite work in a standalone manner, as following the proof sketches for Theorems 4.1 and 4.2 requires some definitions from the supplementary material (eg GOOD).

On page 2, there is a stretch of text that may need fixing "connect with probability $\gamma$. network has recently".

On page 4: "CN(i,j) = {u | i ~ u ~ v}".

I think the "v" should be a "j".

Summary: I am not qualified to pass expert judgement, but this work seemed to be potentially useful (in terms of speed), with some rigorous results on the applicability of a proposed link prediction heuristic.

Parts of the manuscript are not quite self contained (need to refer to supplementary material).

Submitted by Assigned_Reviewer_2

This paper studies the heuristic of common neighbors in link prediction and local clustering around a query node under the stochastic blockmodel setting. The authors show that theoretically, the common neighbors heuristic can extract clusters w.h.p. when the graph is dense enough, and can also succeed in sparser graphs with the addition of a cleaning step.

The quality of this paper is generally good. For clarity it is OK, with some parts difficult to understand. In the end the authors compare SCAN and SCAN+CLEAN with state-of-the-art algorithms on real world data sets, and it seems SCAN and SCAN+CLEAN still perform worse than Spectral Clustering and Katz. It would be good if the authors could discuss about why. Maybe the real world networks do not satisfy the assumptions of the presented theoretical results very well. And it would be good if the authors could discuss, analyze and experiment with those aspects.
Summary: This paper proposes a very interesting study. Overall it looks reasonable and convincing.

Submitted by Assigned_Reviewer_3

Spectral clustering is a standard tool for identifying link prediction. However, it is expensive for large graphs. A popular heuristic is to consider common neighbors, which is a fast and performs well in practise. This paper proves the consistency of the common neighbor heuristic for different classes of graphs. The authors also propose a cleaning step that improves the quality for sparse graphs. The approach is studied empirically on simulated and real world data.

This problem is very relevant and the theoretical analysis seems to be mathematically sound. However, the practical relevance remains unclear, since the experiments fail to explore all strengths and weaknesses: - The proposed algorithm depends on a number of unknown parameters. The authors fail to explain the significance of the link probability of the clusters and the cluster sizes. It is not clear how the algorithm performs if these parameters are not optimally chosen.

- Depending on if the graph is semi-sparse or not, the authors propose to conduct a cleaning step and Figure 1 shows that in that scenario it improves recall and precision. Given that it is hard to check the sparsity in practise, it is also important to show how the cleaning step affects the quality when it is not appropriate. - Equal cluster sizes (\pi = n/K) seems to be a strong assumption. You mentioned that this assumption could be easily relaxed. Can you give an idea of how that would influence your analysis? I assume that relaxing this assumption introduces more parameters that are unknown and we need to make this assumption anyhow. I would expect also an experiment showing the quality if the cluster sizes are skewed.

Section 3 and 4 are very confusing and a clear structure is missing; they introducing and mixing different concepts where it is unclear how they relate together and for what they are needed. E.g., - the formular (line 182-187) are detached from the text - how do the sections "current approaches", "common neighbors", "dependence", "dependence",... relate to each other?

Minor: - Line 97 seems to be a copy&paste error (see Line 126) - Line 163: "eigenvectors of the graph" -> "eigenvectors of the graph Laplacian matrix" - please write out w.h.p (Line 23) and WLOG (Line 222) - Line 214: "when average degree" -> "when the average degree"

After rebuttal: ---------------- Thank you for addressing my concerns about the algorithm's parameters, when the cleaning step is needed, and the number of clusters. However, I find Section 3 and 4 still confusing. I adapted my score accordingly and would like to encourage the authors to clarify the structure and elaborate these issues in the paper.
Summary: + solid theoretical analysis of a popular approach - presentation needs to be improved - experiments fail to explore all strengths and weaknesses - practical relevance questionable

Submitted by Assigned_Reviewer_4

The article provides some theoretical justification for the common neighbors heuristic for link prediction. Under the stochastic blockmodel, the paper shows for a dense graph, the heuristic successfully recovers the clusters. When the graph is sparser, the heuristic plus a cleaning step successfully recovers the clusters.

The problem is equivalent to clustering a realization of the stochastic blockmodel with weak dependencies among edges. To the best of my (limited) knowledge of the area, such a model has not be studied. However, known proof techniques are readily adapted to the setting. Overall, the paper is well-written. If the nearest-neighbor heuristic is commonly used in practice, but unjustified theoretically, then this paper is a worthwhile addition to the literature.
Summary: The article provides some theoretical justification for the common neighbors heuristic for link prediction. If the nearest-neighbor heuristic is commonly used in practice, but unjustified theoretically, then the theory in this paper is a worthwhile addition to the literature.

Author Feedback
Author rebuttal: We thank the reviewers for their comments, and will incorporate their comments
into our text.

Equal cluster sizes:
The equal cluster sizes were used for ease of exposition. We can easily extend
this to the more general setting. Not surprisingly, it requires a higher
separation between alpha and gamma. In particular, line 266 would be replaced by an analogous condition with (\pi_min\alpha-\pi_max \gamma) in place of (\alpha-\gamma), where the smallest cluster is of size n\pi_min and the largest is of size n\pi_max. We also redid our experiments where the smallest cluster was one third in size of the largest. The results are qualitatively similar, the only difference being that the precision and recall curves reach 1 (the perfect score) somewhat slower
than with the equal cluster sizes setting.

Algorithm parameters:
We stated our algorithms so as to simplify the theoretical analysis. While the
algorithm statements need the knowledge of the threshold t_n, this is not
required in practice. Instead we simply do 2-means on the common neighbors
vector (line 356-357). In fact, implementation wise we do not need any
parameters, not even the knowledge of k, which is needed for most other
clustering methods.

When is CLEAN needed:
Theoretically, one may apply CLEAN to both semi-sparse and semi-dense graphs;
in the latter case, it simply adds more computation without improving
guarantees. This was confirmed by experiments where we always do the CLEAN
step. We saw that this generally only improves the results.

Comparison to Katz and spectral clustering:
We presented the real data experiments to show that even though we cannot show anything theoretically for very sparse networks, in practice, out algorithms perform surprisingly well on real sparse networks.

The primary characteristic that differentiates common neighbors from all other link prediction methods is speed. Indeed, while both Katz and Spectral clustering are in some cases more accurate, they are hard to scale to large graphs: Katz requires a matrix inversion (inv(I-beta*A) for some small value beta), while spectral requires an eigen-decomposition. Both of these are impractical on large networks. Common neighbors, on the other hand, still applies.